# CRISPR Approaches for the Diagnosis of Human Diseases

**DOI:** 10.3390/ijms23031757

**Published:** 2022-02-03

**Authors:** Pilar Puig-Serra, Maria Cruz Casado-Rosas, Marta Martinez-Lage, Beatriz Olalla-Sastre, Alejandro Alonso-Yanez, Raul Torres-Ruiz, Sandra Rodriguez-Perales

**Affiliations:** 1Human Cancer Genetics Program, Centro Nacional de Investigaciones Oncologicas (CNIO), Molecular Cytogenetics & Genome Editing Unit, Melchor Fernandez Almagro, 3, 28029 Madrid, Spain; ppuig@cnio.es (P.P.-S.); mccasado@cnio.es (M.C.C.-R.); mmlage@cnio.es (M.M.-L.); beatrizolalla1@gmail.com (B.O.-S.); aayanez@ext.cnio.es (A.A.-Y.); 2Centro de Investigacion Energeticas Medioambientales y Tecnologicas (CIEMAT), Advanced Therapies Unit, Hematopoietic Innovative Therapies Division, Instituto de Investigacion Sanitaria Fundacion Jimenez Diaz (IIS-FJD, UAM), 28040 Madrid, Spain

**Keywords:** CRISPR, Cas13, diagnostic, human diseases

## Abstract

CRISPR/Cas is a prokaryotic self-defense system, widely known for its use as a gene-editing tool. Because of their high specificity to detect DNA and RNA sequences, different CRISPR systems have been adapted for nucleic acid detection. CRISPR detection technologies differ highly among them, since they are based on four of the six major subtypes of CRISPR systems. In just 5 years, the CRISPR diagnostic field has rapidly expanded, growing from a set of specific molecular biology discoveries to multiple FDA-authorized COVID-19 tests and the establishment of several companies. CRISPR-based detection methods are coupled with pre-existing preamplification and readout technologies, achieving sensitivity and reproducibility comparable to the current gold standard nucleic acid detection methods. Moreover, they are very versatile, can be easily implemented to detect emerging pathogens and new clinically relevant mutations, and offer multiplexing capability. The advantages of the CRISPR-based diagnostic approaches are a short sample-to-answer time and no requirement of laboratory settings; they are also much more affordable than current nucleic acid detection procedures. In this review, we summarize the applications and development trends of the CRISPR/Cas13 system in the identification of particular pathogens and mutations and discuss the challenges and future prospects of CRISPR-based diagnostic platforms in biomedicine.

## 1. Introduction

The development of genetic and pathogenic tests has improved healthcare outcomes, as they help to diagnose, monitor and update disease information and reduce negative outcomes, such as adverse drug reactions. In the last decade, molecular diagnostics, which investigate molecules, such as proteins, DNA, and RNA, in a tissue or fluid to identify a disease, have experienced significant development and growth. Infectious diseases, genetics, pharmacogenomics, and oncology are just a few of the topics covered by molecular diagnostics. In this regard, the current SARS-CoV-2 outbreak, the evolution or re-appearance of infectious diseases and the worldwide antibiotic resistance have evidenced the fundamental value that diagnostic processes have in our society [1], and reveal the need for innovative point-of-care (POC) detection methods with high sensitivity and specificity. Nucleic acid detection is a key molecular diagnostic procedure that has been steadily increasing over the last few decades [2]. The most common nucleic acid detection kits are based on quantitative polymerase chain reaction (qPCR), coupled to reverse transcription in the case of RNA, although other options, such as isothermal amplification and next-generation sequencing diagnostics, are also used in routine clinical practice. Quantitative polymerase chain reaction (qPCR) or sequencing have been extensively employed and are generally utilized in clinical laboratories for the detection of nucleic acids. Because of its versatility, robustness, and sensitivity, PCR is the most widely used method for detecting DNA and RNA biomarkers. To obtain reliable and reproducible results, the gold standard PCR approach of accurate quantification by real-time qPCR relies on optimizing numerous processes, such as DNA or RNA extraction, (RNA integrity control and cDNA synthesis in the case of RNA), primer design, amplicon detection, and data normalization [3,4]. Even though isothermal nucleic acid amplification eliminates the requirement for heat cyclers, non-specific amplification can lead to reduced detection specificity [5]. For that reason, there is a need for methods that integrate the ease of use and cost efficiency of isothermal amplification with the diagnostic accuracy of PCR. Single-nucleotide specificity is required for next-generation diagnostics, as it is critical for genotyping, detecting cancer mutations and mutations that confer resistance to antibiotics, antiviral medicines or cancer drugs, and the identification of most virulent pathogenic bacterial or viral variants and strains. However, most of the existing techniques normally require sophisticated equipment and tedious sample/reagent processing, which calls for well-established laboratories with dedicated instruments and well-trained operators and have trade-offs in performance metrics such as sensitivity and specificity. In this respect, the CRISPR/Cas system can open a new window of possibilities for genetic diagnostics.

Clustered Regularly Interspaced Short Palindromic Repeats (CRISPRs) together with CRISPR-associated (Cas) proteins are a prokaryotic adaptive immune system present in various archaea and bacteria that targets the genomes of pathogens. Since its identification, functional description [6,7,8], and subsequent design to make it programmable and efficient to edit the genome [9], CRISPR/Cas has become a revolution in the biotechnology field. Genome editing, gene therapy, epigenetic regulation, and library construction are only a few of the uses of CRISPR [10,11,12,13]. The fact that nucleic acids are effective biomarkers for many diseases, together with the ability of the CRISPR/Cas system to recognize specific nucleotide sequences, and the discovery of new Cas protein effectors (Class 2, type V; Cas12 and Cas13) has opened the door to new tools that offer cost-effective and portable diagnostics through nucleic acid screening (Figure 1).

## 2. Classification of CRISPR/Cas Systems

Archaeal and bacterial CRISPR/Cas natural defense systems show a wide diversity in architecture and mechanisms of action, and new variants are still being reported [33]. Between 2011 and 2020 [34,35,36,37], Koonin and collaborators proposed and improved the currently accepted classification of CRISPR/Cas systems, based on phylogenetic relationships. According to the last update, CRISPR/Cas class 1 systems (which include types I, III and IV) use an effector complex composed of multiple Cas proteins. In contrast, class 2 systems (types II, V and VI) present a single, larger, multi-domain effector Cas protein [18,20,38].

In the mid-to-late 2010s, class 2 systems were only represented by Cas9. Its simplicity, in comparison to multi-protein class 1 systems, encouraged computational efforts to uncover new CRISPR effectors [18]. A variety of class 2 CRISPR/Cas systems were discovered [18,38]. In the case of the class 2 type II systems, the canonical effector protein is Cas9, which targets double-stranded DNA (dsDNA). Unlike other nuclease effectors, Cas9 requires the presence of a trans-activating CRISPR/Cas RNA (tracrRNA) that enables the interaction between the protein and a single guide RNA (sgRNA), which contains a sequence complementary to the target region. Cas9 requires a protospacer adjacent motif (PAM) downstream of the target sequence to produce a blunt-ended double-strand break (DSB) [39,40]. Two extensively characterized orthologs of the Class 2 type V CRISPR/Cas systems are Cas12a and Cas12b. Both are capable of recognizing and cleaving target sequences near a PAM site in dsDNA, just as Cas9. However, Cas12a does not require a tracrRNA and an additional nuclease to process the crRNA, since it can mediate the maturation of its own crRNA [41]. Additionally, Cas12 effectors produce cohesive-ended DSBs [42]. While for most prokaryotic CRISPR/Cas systems target DNA, late in the 2000s, Terns and collaborators reported for the first time a CRISPR/Cas effector complex that recognizes viral invader RNA and cleaves it [43]. The class 2 type VI CRISPR/Cas systems rely on the presence of Cas13 protein effectors that contain two catalytic domains responsible for cleaving single-stranded RNA (ssRNA) target sequences. The nucleolytic activity of Cas13 is triggered when the complex binds the targeted ssRNA and the Protospacer Flanking Sequence (PFS) is recognized at the crRNA binding site [44]. This PFS region is the equivalent of the PAM sequence for RNA-target recognition and varies between the multiple subtypes of CRISPR/Cas13, thus enabling differential nucleic acid targeting [40].

## 3. Applications of CRISPR/Cas Systems in Molecular Diagnostics

The combination of Cas9 activity with other techniques, such as nucleic acid amplification, is being used for the detection of specific DNA and RNA sequences, thus applying it as a biosensing system for diagnostic purposes. The applications of Cas9-based biosensing systems can be classified into two categories: those leveraging the specific target cleavage activity of the Cas9 effector, and those combining the target-specific binding activity of nuclease-deficient Cas9 (dCas9) with split signal-transducing proteins [45]; such biosensors can be applied to genotyping pathogens and discriminating single nucleotide polymorphisms (SNPs). In 2016, Pardee and collaborators successfully combined Cas9 cleavage activity with nucleic acid amplification technologies, such as nucleic acid sequence-based amplification (NASBA), and an isothermal amplification technique based on CRISPR cleavage (NASBACC). This combination was used to distinguish with single base resolution closely related RNA Zika virus strains in vitro using toehold switch RNA sensors and a freeze-dried, paper-based platform [19]. This approach is based on the idea that only in the presence of the Zika virus RNA strand the toehold unfolds, revealing a ribosome binding site and resulting in the translation of proteins that induce a visible color shift. To detect this color shift in samples with clinically relevant Zika RNA levels, toehold switches are paired with an isothermal RNA amplification technique along with a system dried onto paper discs [19]. In 2018, Xing and collaborators developed a novel CRISPR/Cas9-triggered isothermal exponential amplification reaction (CAS-EXPAR) strategy for site-specific and rapid fluorescent nucleic acid detection. This method has been applied for the detection of short microRNAs (miRNAs) [24]. FLASH is another method based on CRISPR/Cas9 developed to detect drug-resistant pathogens [46]. This technique uses an array of sgRNAs and Cas9 proteins that cleave the genes of interest, generating small fragments that are suitable for Illumina sequencing. This has been applied in *Staphylococcus aureus* and *Plasmodium falciparum* [31]. In 2020, Wang and colleagues merged a lateral flow nucleic acid assay with the CRISPR/Cas9 system in a kit named CASLFA (CRISPR/Cas9-mediated lateral flow nucleic acid assay) that was used to detect African swine fever virus [47]. The utilization of a catalytically inactive form of Cas9 that selectively binds to target DNA, but does not cleave it, led to the development of FELUDA (FNCAS9 Editor-Linked Uniform Detection Assay) used for SARS-CoV-2 molecular testing [48].

One of the main features of the Cas12 effector that distinguishes it from Cas9 is the collateral cleavage activity of the former. Once the complex Cas12/crRNA/target DNA is established, the indiscriminate (non-specific) cleavage of any collateral single-stranded DNA (ssDNA) is induced by the catalytic domain of the nuclease. This peculiar activity has been exploited to create biosensing systems that merge Cas12 effectors with amplification methods to enable rapid and specific detection of pathogen DNA samples [41,49]. The diagnostics platforms using Cas12a utilize ssDNA probes, which implies more stability. After the isothermal amplification of the target viral gene, virus-specific crRNAs, Cas12 effectors, and ssDNA probes are introduced into the tube and incubated. Cas12a is activated when it attaches to target dsDNA and cleaves the amplified DNA as well as probes that emit fluorescence. Fluorescence can be monitored with a fluorometer or by using paper-based detection techniques. One-HOur Low-cost Multipurpose highly Efficient System (HOLMES) and DNA Endonuclease Targeted CRISPR Trans Reporter (DETECTR) are two major CRISPR/Cas12-based diagnostic systems, which have been applied worldwide [50,51]. HOLMES advanced to HOLMESv2, which uses Cas12b instead of Cas12a to detect SNPs and different viruses, such as the Japanese encephalitis virus (JEV) [52]. DETECTR has been used recently for the CRISPR/Cas12a-based detection of SARS-CoV-2 [30].

The Cas13 effector is widely utilized to produce RNA knockdown models, but it has also been applied for efficient and specific RNA detection as a biosensing system. An important feature of this effector comprises a collateral RNase cleavage activity that is triggered after the target sequence is cleaved by the Cas13/crRNA/target RNA complex. Four subtypes of the Class 2 type VI Cas13 nuclease have been identified so far; although they differ in size and sequence, they all contain two catalytic domains that are responsible for cleaving single-stranded RNA (ssRNA) target sequences [53]. Cas13 is a unique RNA-nucleolytic RNA-guided protein that once activated by binding to the target ssRNA, cleaves nearby RNAs in trans. Regarding the formation of the effector complex, the Cas13 protein processes its own crRNA, but other host nucleases participate in crRNA maturation [44]. The Cas13 protein does not require a PAM sequence, but some subtypes require a single base-specific protospacer flanking site (PFS) for target RNA recognition that varies between the multiple subtypes of CRISPR/Cas13, thus enabling differential nucleic acid targeting [40]. In the manner of HOLMES, Cas13-based detection systems require isothermal amplification of the target genome, crRNAs, and fluorescent ssRNA probes. The Cas13-based SHERLOCK (specific high-sensitivity enzymatic reporter unlocking) system is the first demonstration of an indiscriminate cleavage diagnostic CRISPR/Cas system. When indiscriminate cleavage activity occurs by recognition of the target RNA by the Cas13-crRNA complex, the ssRNA of the reporter molecule is cleaved, separating the fluorescent molecule from the ssRNA bound quencher, resulting in fluorescence [22]. It has also been demonstrated that the reporter may be modified for use on lateral flow assay strips by substituting two proteins/antigens that can be bound by antibodies in the capture lines of the strip for the dye–quencher pair [25,26]. The Heating Unextracted Diagnostic Samples to Obliterate Nucleases (HUDSON) methodology was developed to identify viral genetic material from body fluids, such as urine, blood and its isolates, and saliva, making the SHERLOCK procedure even more efficient [26]. HUDSON protocol researchers found that conserved regions within the genetic material of these viruses can be identified using universal-flavivirus Recombinase Polymerase Amplification (RPA) primers, as well as crRNAs specific to a given viral species [26]. SHERLOCK and HUDSON protocols can be applied to any virus, but previous testing focused on the diagnosis of flaviviruses, such as Zika, Dengue, West Nile, and yellow fever viruses [25,26].

Each of the previously described CRISPR-based nucleic acid detection systems (Figure 2) has its own pros and cons. The CRISPR/Cas9-based diagnostic tools do not provide a strong, specific signal when target nucleic acids are present in the sample, and have been linked to a PCR to pre-amplify nucleic acids, a method that limits the range of CRISPR diagnostic possibilities. On the other hand, CRISPR/Cas12 and/Cas13-molecular diagnostic kits can be utilized in totally isothermal circumstances and require little equipment and training of people; they have already been used for colorimetric and visual detection of pathogens on lateral flow strips, allowing these tests to be used for mass screening and quick diagnosis in almost any geographic location [54]. Given the great progress in its applicability during the last few years, we will now focus on the technical aspects and applications of the CRISPR/cas13 system.

## 4. The CRISPR/Cas13 System

### 4.1. Cas13 Family Members

As described above, Cas13 proteins are programmable RNA-guided nucleases that target single-stranded RNA (ssRNA) substrates [41,55]. Although some class 1 CRISPR systems are able to target RNA, class 2 type VI systems are the only known single effector nucleases that are RNA-specific [23,56]. Four Cas13 subtypes have been identified thus far, named a–d. In all of them, two R-x_4_-H motifs, characteristic of higher eukaryotes and prokaryotes nucleotide-binding (HEPN) domains, are present. These are the sites involved in RNA cleavage and the common feature shared by all of them, since the amino acid sequences are significantly distinct [18,38,55,56]. Cas13a (formerly C2c2) is the RNase involved in the VI-A subtype CRISPR/Cas systems. It was characterized in the bacterium *Leptotrichia shahii*, and its interference activity against phage RNA was demonstrated in 2016 [20]. In the class 2 subtype VI-B1 and VI-B2 CRISPR/Cas systems, Cas13b (formerly C2c6) is regulated by Csx27 and Csx28, respectively [18,38]. In mammalian cells, the PspCas13b effector, a Cas13b ortholog from *Prevotella* sp. P5-125, has demonstrated highly effective and specific RNA knockdown [57]. Another Cas13 family member is Cas13c (C2c7), an effector RNase in VI-C subtype systems. More recently, Cas13d was described as the effector protein involved in subtype VI-D CRISPR/Cas systems [58,59]. This effector protein is approximately 100 kDa in size, 20–30% smaller than the rest of Cas13 family members, which have a molecular weight of around 130 kDa [23,59]. It has been reported to achieve effective and reliable knockdown of several endogenous transcripts [57,58,60].

### 4.2. crRNA Architecture and Processing Mechanism

All CRISPR/Cas systems require three stages to confer adaptive protection against prokaryotes: (i) adaptation, (ii) expression and processing, and (iii) interference [59,61]. The acquisition of new spacers between the repeats within the CRISPR array occurs in the first stage. In most CRISPR/Cas systems, the proteins Cas1 and Cas2 are involved in this mechanism, whereas they are absent in almost every type VI system. It remains unclear how these latter systems acquire spacers [62]. During the expression and processing stage, the CRISPR array is transcribed into a single precursor CRISPR RNA (pre-crRNA). This long transcript contains alternating repeat and spacer sequences and must be processed by splitting into individual, short crRNAs. In type VI CRISPR systems, this task is carried out by Cas13 itself, without the assistance of other host factors [18,55]. A 28–30 nucleotide spacer and a 28–36 nucleotide direct repeat make up the mature form of crRNA. The hairpin structure derived from the direct repeat is placed in the 5′ end of the crRNA in all CRISPR VI subtypes [20,58,59], except for VI-B, which presents it in the 3′ end [38]. Finally, during the interference step, a single Cas13 nuclease binds to a mature crRNA through said hairpin, generating a ribonucleoprotein (RNP) complex. When the crRNA, acting as a guide, binds to target RNA via base complementarity, the RNP complex undergoes a conformational change. As a result, the two Cas13 HEPN domains converge to form a single catalytic site, where the RNA is cleaved [56,63]. Hence, the Cas13 effector is responsible for both maturation and interference activities in type VI CRISPR systems [56]. Interestingly, this catalytic site is slightly set back from the crRNA-targeted RNA duplex [20]. Consequently, Cas13 has a collateral, non-specific trans-activity: after its activation, both target RNA and any surrounding ssRNAs are degraded indiscriminately [20,56]. The use of type VI Cas effectors as tools in diagnostic assays is based precisely on this indirect effect [22,25,64].

### 4.3. The Protospacer Flanking Sequence

Type VI systems have a security mechanism that prevents the cleavage activity from being triggered by the own RNA of the host. HEPN domains must detect the so-called Protospacer Flanking Sequence (PFS) in close proximity to the target protospacer sequence in order for the catalytic site to be activated [65]. Cas13a must recognize a non-G PFS in nucleotides 13–24 downstream of the crRNA binding site [20,66]. Cas13b needs a double-sided PFS: a downstream non-C PFS as well as a NAN/NNA (being N any nucleotide) sequence in nucleotides 12–26 upstream [38]. PFS recognition is not required for the activation of Cas13d in VI-D systems [23,59]. Nevertheless, PFS dependence constraints for Cas13 nucleases are less rigid than PAM requirements for Cas9 nucleases and vary across orthologs [38,44,55,59,67,68], making it a more versatile targeting method.

## 5. Nucleic Acids Detection with CRISPR/Cas13

Many applications in human health and biotechnology rely on rapid nucleic acid detection, such as the identification of infectious agents or disease-associated circulating DNA or RNA. The CRISPR/Cas13 system has been developed for the selective, fast, sensitive, and portable detection of nucleic acids [69,70]. These methods rely on the nonspecific Cas13 endonucleases that bind to a specific target via programmable crRNAs. The Cas13 enzyme produces a specific and sensitive indicator of the presence or quantity of a nucleic acid by combining the programmable specificity of Cas13 with a reporter molecule that is triggered upon target identification. In this paper, we detail the main steps included in the Cas13 protocol for nucleic acid detection (Figure 3).

### 5.1. Sample Pre-Treatment: Nucleic Acid Extraction

A key piece in achieving usable Cas13-based molecular diagnostics methods is the simplification of the sample preparation. Important efforts have been made to isolate nucleic acids and remove nucleases rapidly and without equipment, ensuring that the reagents used in these procedures do not interfere with any subsequent preamplification steps or with Cas13-based target detection. According to the original SHERLOCK methodology, boiled saliva pellets can be processed directly, using a lysis combination of 0.2% Triton X-100 with phosphate-buffered saline and heating at 95 °C for 5 min, for human genotyping [22,25]. To increase the variety of sample types, the HUDSON protocol (Heating Unextracted Diagnostic Samples to Obliterate Nucleases) was designed to inactivate nucleases and viruses via heat and chemical reduction using 100 mM tris(2-carboxyethyl)phosphine hydrochloride and 1nM ethylenediaminetetraacetic acid (EDTA), varying incubation time and temperature depending on the sample and target type. With HUDSON, Zika and Dengue virus were detected with LwaCas13a (with RPA amplification) from blood, plasma, serum, saliva and urine [26]. HUDSON has been optimized for SARS-CoV-2 RNA detection [71]. However, it was insufficient for the detection of *Plasmodium* sp., but a buffer with stronger chelating agents restored sensitivity [72].

### 5.2. Nucleic Acid Amplification

Since most clinical settings require the ability to detect nucleic acid concentrations below the picomolar range [73], the majority of CRISPR/Cas13-based diagnostics published to date depend on target preamplification. Although PCR is used in some investigations [27,51], isothermal amplification techniques are more suitable for POC application. There are some examples of CRISPR systems coupled to isothermal enzyme-free amplification reactions, such as hybridization chain reaction (HCR) [74], entropy-driven circuit (EDC) [75], and catalytic hairpin assembly (CHA) [76]. However, in fact, most CRISPR diagnostics use enzyme-assisted amplification methods, including rolling circle amplification (RCA) [77,78], strand displacement amplification (SDA) [79], loop-mediated isothermal amplification (LAMP) [52,80], nucleic acid sequence-based amplification (NASBA) [19,22], exponential amplification reaction (EXPAR) [24] and allosteric probe-initiated catalysis (APC) [81].

In Cas13-based diagnostics, the most commonly used isothermal amplification method is RPA [25,71,82,83], because it functions at near-ambient temperature (37–42 °C), is easily combinable with reverse transcription for RNA samples, and primers are easy to design. Initially, SHERLOCK used RPA for preamplification, which showed higher sensitivity than NASBA [22]. In traditional RPA, non-specific amplification is a problem, but this is circumvented by the downstream crRNA-based target detection.

### 5.3. Readout Options

Fast and low-cost readouts are the most suitable for field applications and several ways for measuring and visualizing the activation of Cas13 enzyme upon target detection have been used, such as fluorescence, colorimetry, paper-based lateral flow and electrochemistry.

For fluorescence-based readouts, the reporter RNA carries a fluorophore and a quencher in close proximity; they are spatially separated from each other by the trans-RNase activity of the target-activated Cas13, and thus emit fluorescence only when the target RNA is present. This emission can be read on conventional plate readers [22,25,84,85], on low-cost and portable devices [86], and even with the naked eye under blue light [71].

Another Cas13 readout method is colorimetric visualization, for instance with gold nanoparticles that are linked through ssRNA oligonucleotides: when Cas13 is activated by the target RNA, it cleaves the ssRNA oligonucleotides, the gold nanoparticles are dispersed, and the purple solution turns red [82]. Visual Cas13-readouts can also be obtained using a turbidity-based assay based on the liquid–liquid phase separation of nucleic acids and positively charged polyelectrolytes. On target recognition, collateral cleavage results in the breakdown of nucleic acid polymers. After complementation with polycations, the solution either remains clear when cleavage has happened or becomes turbid when no target-triggered cleavage has occurred [87]. In another study, Cas13a collaterally cleaves the uracil ribonucleotide (rU)-bearing pre-primer upon recognition of target RNA. The remaining 5′-DNA fragment of the pre-primer can then initiate DNA polymerase-mediated rolling circle amplification (RCA) after 3′-end repair by T4 polynucleotide kinase (T4 PNK). The amplification generates a long G-rich repeat sequence, which can form a tandem G-quadruplex that acts as a horseradish peroxidase mimicking DNAzyme to catalyze the oxidation of the 2,2′-azino-bis(3-ethylbenzothiazoline-6-sulfonic acid) diammonium salt (ABTS^2−^) in the presence of hemin, generating a visible color change [77].

Paper-based lateral-flow immunochromatography technology offers a simplified representation of the collateral cleavage activity of Cas13. The most widely used commercially available system (Millenia 1T) consists of a strip with two capture lines [25,26,71,83]. One contains streptavidin (closer to the sample pad area); the other one bears species-specific secondary antibodies (farther from the sample pad area); and in the sample pad area, anti-fluorescein isothiocyanate (FITC) antibodies coupled to gold nanoparticles (GNPs) are present. Reporter RNA molecules carry biotin and fluorescein amidite (FAM). In the absence of the target, uncut reporter RNA molecules with anti-FITC GNPs attached to the FAM are retained only in the zone with streptavidin, and this is visualized as one band on the strip. When the target is present, the biotin is separated from the FAM molecules by Cas13-enzyme-mediated collateral cleavage, allowing the anti-FITC GNPs bound to FAM to travel farther on the strip to generate a second visible band at the species-specific antibody-capture line.

Diagnostics based on Cas13 with electrochemical readouts are also used. In one study, RNA reporters carry FAM and biotin, and the sensor surface has streptavidin fixed. The binding of glucose-oxidase-coupled anti-FAM antibodies to uncut RNA-reporter molecules immobilized by biotin–streptavidin binding on the sensor catalyzes the oxidation of glucose when there is no Cas13 collateral cleavage. Using an electrochemical cell with a platinum working electrode, a platinum counter electrode, and a silver/silver chloride reference electrode, the produced H_2_O_2_ is then amperometrically measured [88]. In another example of electrochemical readout, the presence of a target miRNA activates the trans-cleavage activity of a CRISPR/Cas13a system, leading to the release of an initiator of CHA to generate amplified electrochemical signals [89]. A portable electrochemiluminescence chip was also used to show Cas13a-based electrochemical detection of miRNAs. The initiation of an exponential isothermal amplification is triggered by the collateral cleavage of Cas13 in the presence of the target. The amplified products form a complex with [Ru(phenanthroline)_2_dipyridophenazine]^2+^, and when the complex is oxidized at the anode of a bipolar electrode, an electrochemiluminescent signal proportional to the concentration of the target miRNA is created [90].

### 5.4. Amplification-Free CRISPR/Cas13-Diagnostics

Given that the Cas13 enzyme is highly specific, it distinguishes single base mismatches, the preamplification step present in most of the Cas13-based methods is aimed at improving the detection sensitivity of these tests. There are three types of alternatives for the amplification step: increase of target concentration by the reduction of the reaction volume, electrochemical biosensors to improve the sensitivity of the detection, and amplification of the detection signal with other Cas proteins.

The RNA target concentration required to trigger the trans-cleavage activity of Cas13 is around 50 picomolar [22]. By simply reducing the reaction volume, the target concentration increases, improving the limit of detection of an amplification-free assay. Tian and collaborators confined the Cas13 detection reaction in cell-sized reactors to increase the target and reporter concentrations through droplet microfluidics. In comparison with the normal microliter volumes, their picoliter-sized Cas13a system showed a more than 10,000-fold enhancement in detection sensitivity and enabled absolute target RNA quantification. They demonstrated its applicability by detecting microRNAs, 16S rRNAs, and SARS-CoV-2 RNA [91]. Similarly, Shinoda and collaborators, by combining Cas13 detection and microchamber-array technology, created a platform named SATORI, which enables the detection of target RNA at concentrations as low as 5 femtomolar in 5 min. Furthermore, they simultaneously used multiple different guide RNAs to enhance the sensitivity, proving their test applicability with the detection of the SARS-CoV-2 N-gene RNA [92]. Both examples make use of a fluorescence-based readout.

As described above, Cas13 detection can also be monitored with an electrochemical readout, thus increasing the limit of detection of Cas13 and avoiding the use of a preamplification step. Brunch’s multiplexing biosensor described above can be used to detect with 10 picomolar sensitivity, with a readout time of 9 min and an overall process time of less than 4 h [88]. The limit of detection for target RNAs can be improved to 50 attomolar by combining this electrochemical biosensor with a dual signal amplification strategy that includes both Cas13a and a catalytic hairpin DNA circuit, with a readout time of 6 min and an overall process time of 36 min, without the need of target RNA amplification [93].

The weak Cas13 trans-cleavage signal can be increased by Cas-based cascade amplification to provide amplification-free diagnostics. Csm6 RNA endonuclease has been employed for signal amplification by Gootenberg and collaborators, who obtained a 3.5-fold increase in signal sensitivity. Cas13a is triggered to trans-cleave a pre-activator sequence in the presence of target RNA, generating mature activators for Csm6, which then cleaves the RNA reporter creating an amplified fluorescent signal [25]. Recently, this system was further optimized in the FIND-IT system to detect SARS-CoV-2 RNA, by using a chemically stabilized activator. Furthermore, the FIND-IT system can be utilized to diagnose clinical samples quickly, with a diagnostic sensitivity of up to 33 RT-qPCR-derived cycle threshold (Ct) values and detection times of less than 40 min [94]. Cas13a and Cas12f (formerly known as Cas14a) have also been coupled to create the cas-CRISPR system, in which Cas13a trans-cleaved reporter molecules act as activators for Cas12f and further activate Cas12f-mediated trans-cleavage of its own reporters, resulting in amplified output signals. The cas-CRISPR system achieves a detection limit of 1.33 femtomolar, which is 1000 times more sensitive than Cas13a alone, and single-base resolution for miR-17 detection [85].

In addition to the approaches outlined above, there is another alternative to improve detection sensitivity, through optimizing the reaction conditions and designing more than one crRNA for a target nucleic acid sequence. By combining crRNAs, the sensitivity for direct detection of SARS-CoV-2 was enhanced to a sensitivity of ∼100 copies/μL in less than 30 min of measurement time. Moreover, viral load can be directly quantified using enzyme kinetics and the result can be read with a mobile phone microscope developed in the study [95].

### 5.5. Specificity and Sensitivity

A major advantage of Cas13-based diagnostics is its single-nucleotide specificity, which allows the detection of point mutations. In fact, Cas13-based technology has been used to detect specific strains of Zika and Dengue virus, human SNPs, as well as mutations in the *EGFR* (epidermal-growth-factor-receptor), *BRAF* and *APC* (adenomatous polyposis coli) genes [22,25]. The single-nucleotide specificity of the CRISPR/Cas13 system has also been used to recognize miRNAs that differ by only one base, with the added difficulty of their small size [77,84,85,90].

The great sensitivity of Cas13-based detection technology is another cornerstone of its applicability in clinical diagnostics. The limit of detection of CRISPR/Cas13-mediated nucleic acid detection, with a target amplification reaction included, is attomolar [22,71,83] or even zeptomolar with the combination of an auxiliary CRISPR-associated enzyme, Csm6 [25]. The detection limit of Cas13-based amplification-free detection is usually at the picomolar to femtomolar level [85,90,92,96], but recently several amplification-free CRISPR/Cas13 based biosensors have improved their sensitivity to attomolar [91,93,94,95].

The high sensitivity and specificity of Cas13a has enabled the detection of nucleic acids of interest in clinical samples. For instance, a two-step SHERLOCK-based detection assay of SARS-CoV-2 reached a limit of detection (LoD) of 42 RNA copies per reaction; the sensitivity was 96% and 88% when using fluorescence and lateral-flow readouts, respectively, and both readouts were 100% specific in a set of 154 nasopharyngeal and throat swab samples [97]. Another one-step HUDSON optimized SARS-CoV-2 test achieved a LoD of 10 copies per μL and 100 copies per μL with fluorescent and lateral-flow readouts, respectively. This test was conducted on 50 nasopharyngeal patient samples, and it had a sensitivity of 90% and a specificity of 100% when compared to RT-qPCR [71].

A Cas13-based assay for the detection of opportunistic post-transplantation infections and monitoring transplant rejection also showed great sensitivity and specificity rates. Using RT-qPCR results as reference, the Cas13 assay detected BK polyomavirus DNA in 67 HUDSON-isolated plasma and urine samples with 100% sensitivity and 100% specificity. HUDSON-based processing of cytomegalovirus DNA-containing plasma samples yielded 80% sensitivity and 100% specificity, whereas silica-based membrane DNA purification was required to achieve 100% sensitivity and specificity. Finally, in total RNA isolated from 31 urine cell pellets, CXCL9 mRNA was utilized to monitor acute cellular kidney-transplant rejection, as characterized by renal biopsy, with a sensitivity of 93% and a specificity of 76% [83].

### 5.6. Multiplexing: Multiple Analyte Detection

Early attempts to develop multiplexed CRISPR-based diagnostics took advantage of the cleavage preference of distinct Cas enzymes for diverse nucleic acid reporter sequences, including a multiplexed preamplification step (RPA). This enabled Cas enzymes to detect up to four different nucleic acid targets in a single multiplexed test with attomolar sensitivity, by the combination of four reporter molecules that were specifically targeted by the collateral activity of the following orthologs: PsmCas13b (from *Prevotella* sp. MA2016), LwaCas13a (*Leptotrichia wadeii*), CcaCas13b (*Capnocytophaga canimorsus* Cc5) and AsCas12a (*Acidaminococcus* sp.). The differences between the reporter molecules consist of variation in the composition of the nucleotides in the link of the quenched-fluorescent probe for the Cas13 orthologs and variation in the nature of the nucleic acid structure (DNA vs. RNA) for the Cas12 ortholog [25].

However, with the miniaturization of the reaction volume, the Cas13-based multiplexed assay CARMEN (Combinatorial Arrayed Reactions for Multiplexed Evaluation of Nucleic acids) enables the scalable and simultaneous detection of more than 4500 targets. The inputs of CARMEN–Cas13 —amplified nucleic acid target samples (PCR or RPA) and LwaCas13 detection mixes containing Cas13, crRNA and cleavage reporter— are each combined with a distinct solution-based color code that serves as an optical identifier. Each color-coded solution is emulsified in fluorous oil, creating one nanoliter droplets. The droplets from all samples and detection mixes are pooled and subsequently introduced into a microwell-array chip. Two droplets from the pool are placed at random in each microwell in the array, forming all possible pairwise combinations of amplified target samples and Cas13-detection droplets. The contents of each microwell are determined by identifying the color codes of the droplets using fluorescence microscopy. Droplet pairs confined in each microwell are then merged by exposure to an external electric field. On target detection, the fluorescence emitted by Cas13a-triggered reporter cleavage shows a positive reaction [98].

Another example of Cas13-detection multiplexing is the electrochemical microfluidics-based chip described by Bruch and collaborators. The readout is based on the amperometric detection of H_2_O_2_ produced by the oxidation of glucose when there is no Cas13 collateral cleavage, previously reported by these authors [88]. They designed and implemented different multiplexed versions of their electrochemical microfluidic biosensor, by dividing the channels into subsections, creating four novel chip designs for amplification-free and simultaneous quantification of up to eight miRNAs with Cas13; this system is named CRISPR-Biosensor X [96].

## 6. CRISPR/Cas13 Clinical Applications

Molecular diagnostic tests for genetic or virus-associated disorders are critical for improving healthcare outcomes by making testing quick, accurate, and simple. As previously mentioned, Cas13 is an RNA-guided protein (class 2 type VI) with RNase activity that cleaves adjacent RNAs in a non-specific way once activated [20]. Cas13-based diagnostics has been implemented for the detection of specific nucleic acids characteristic for, e.g., oncogenic mutations and infectious diseases.

### 6.1. Cas13-Based Diagnostics for Infectious Diseases

Initially, the main focus of Cas13 as a diagnostic tool was to be able to detect genetic material associated with pathogens (viruses, mainly). Because of its nature as a diagnostic tool based on RNA detection, RNA viruses are the preferred target for its application, and it has been implemented for the detection of Dengue [19,25,26], Zika [19,22], Ebola [99], SARS-CoV-2 [71,97] and white spot syndrome virus [100,101].

The previously mentioned SHERLOCK-based detection system was implemented at the beginning of the COVID-19 pandemic as a diagnostic tool to accelerate and facilitate population screening [97]. In fact, its two-step version (using RT-LAMP) received emergency use authorization of the CRISPR-based detection of SARS-CoV-2 from the United States Food and Drug Administration (FDA). The assay was able to detect as low as 42 viral RNA copies per reaction with great sensitivity and efficiency. A different Cas13a-based assay combines preamplification and detection reactions into a single test tube and uses HUDSON to speed up viral extraction from saliva and nasal swabs [71]. After process optimization, its performance was comparable with the results obtained by the two-step assay. Even more, a next step level was achieved by the implementation of a Cas13a homologue (LbuCas13a, *Leptotrichia buccalis*) that enables similar results, but without needing a preamplification step [95].

Furthermore, the Cas13-detection system has been used in DNA virus detection, such as Epstein–Barr virus (EBV) [102], BK polyomavirus and cytomegalovirus (CMV) [83]. Because of the DNA nature of those viruses, a modified version of the SHERLOCK protocol was implemented. In the first place, an RNA detectable by the CRISPR/Cas13 is generated by RPA reaction incorporating a T7 promoter embedded into the forward primer. To detect EBV in plasma samples from patients with nasopharyngeal carcinoma, a simplified SHERLOCK protocol was carried out at room temperature. DNA was detectable in 46 out of 48 samples from positive patients and in 0 out of 50 healthy control samples. Results that are in line with those obtained by qPCR. In the BK polyomavirus and CMV study a HUDSON methodology was employed, showing very high sensitivity (100% for BKV and 80% for CMV) and specificity (100% in both cases) in a cohort of 67 urine and plasma samples. qPCR was employed to validate the HUDSON results.

Apart from viruses, the system has been also implemented for detection of different pathogens such as bacteria; in this case, it includes opportunistic pathogens, such as *Pseudomonas aeruginosa* [22] and *Staphylococcus aureus* [22,25], and others that cause different human diseases, such as tuberculosis (*Mycobacterium tuberculosis,* [22]) typhoid fever (*Salmonella enteritidis,* [81]) and Listeriosis (*Listeria monocytogenes,* [24]). The system is also able to detect parasites, such as the one that causes malaria (*Plasmodium falciparum,* [72]).

In comparison to Cas9, Cas13 has been implemented as a more sensitive system for pathogens and virus detection. Cas9-based diagnostics coupled with a pre-amplification step has a LoD that ranges between 1 × 10^−15^ M [19] and 2 × 10^−19^ M [47], while Cas13-based diagnostic has a LoD that range between 3 × 10^−18^ M [80] and 8 × 10^−21^ M [25].

### 6.2. Cas13-Based Diagnostics for Non-Infectious Diseases

Apart from its great applicability in the field of virus diagnostics, Cas13-based diagnostics have been implemented for the detection of non-infectious diseases. For example, in order to be able to detect graft-versus-host disease in kidney transplants, the system is able to detect abnormal levels of human CXCL9 mRNA, an indicator of acute cellular kidney-transplant rejection [83]. Apart from mRNAs, the system is able to detect specific miRNAs in medulloblastoma patients (miR-19b [88]) and breast cancer cell lines (miR-17 [84]).

Finally, one key point that the system has demonstrated since its implementation is its specificity, which permits the discrimination of point mutations (SNPs) and small deletions. That is the case for mutations and deletions in the *EGFR* and *BRAF* genes [25]. The combination of such a specific potential with liquid biopsy sample collection makes cancer diagnosis more affordable and less invasive. It has been demonstrated that cell-free DNA harbors cancer-associated mutations, and this combination has been harnessed for the detection of cancer-related mutations (EGFR-L858R and BRAFV600E) in cell-free DNA samples containing as little as 0.1% mutant alleles [22]. The sensitivity of Cas13-based diagnosis here was similar to that of ddPCR (digital droplet PCR) and qPCR.

In conclusion, combining isothermal amplification with Cas13-based detection enables very sensitive and specific detection of cell-free DNA from liquid biopsy samples, which makes it more affordable and obviates the need for expensive laboratory equipment.

## 7. Concluding Remarks

The CRISPR/Cas systems have revolutionized biotechnology, demonstrating great capability in genome editing, gene therapy, and, more recently, nucleic acid diagnostics. CRISPR-based molecular diagnostics take advantage of the robust and highly specific characteristics of CRISPR/Cas systems to create ultra-sensitive, low-cost, and quick non-laboratory-based detection kits for genetic disorders, viruses, and other pathogens. This vastly improves on existing diagnostic tools, and opens up a slew of new possibilities for tackling important human health issues. We outlined the various platforms based on the CRISPR-associated nucleases Cas9, Cas12, or Cas13 in this review, focusing on Cas13. After incorporating an effective sample preparation unit, which is important for constructing functional sample-to-answer molecular diagnostics, developments using the Cas13 system have shown good progress toward a realistic POC diagnostics. As the globe faces rising pressures on medical systems from antibiotic resistance to viral epidemics, this progress will help to address the growing need for speedy and accurate diagnostic tests in resource-constrained settings.

Several active clinical trials are attempting to diagnose pathogens using CRISPR-based diagnostic tools (ClinicalTrials.gov identifiers: NCT05143593, NCT04535505, NCT04178382, NCT04074369, NCT04535648). SARSCoV-2 detection kits based on Cas12 and Cas13 were recently approved by the FDA for emergency use [103]. As a result, CRISPR-based diagnostic techniques may soon be deployed in clinical practice. CRISPR/Cas13 is a CRISPR/Cas system with biochemical features that make it useful for diagnostics. Cas13-based biosensors are inexpensive, sensitive, selective, and simple to use for detecting DNA, RNA, and proteins in liquid biopsy samples. Despite the fact that we have yet to fully utilize the potential of CRISPR/Cas13-based platforms, this technology represents a revolutionary advance, and future research will reveal the potential and limitations of CRISPR/Cas13.

## Figures and Tables

**Figure 1 ijms-23-01757-f001:**
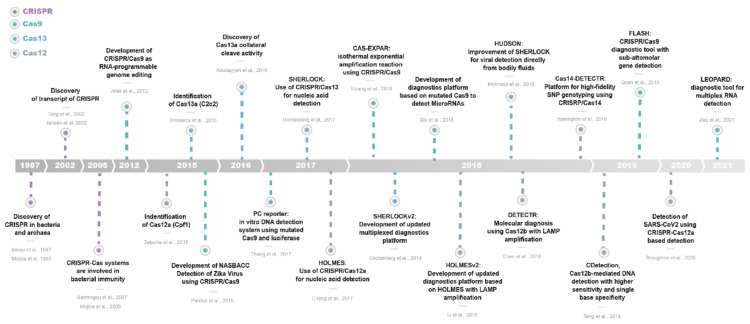
Timeline of milestones of CRISPR/Cas nucleic acid detection-based diagnostic methods [6,7,8,14,15,16,17,18,19,20,21,22,23,24,25,26,27,28,29,30,31,32].

**Figure 2 ijms-23-01757-f002:**
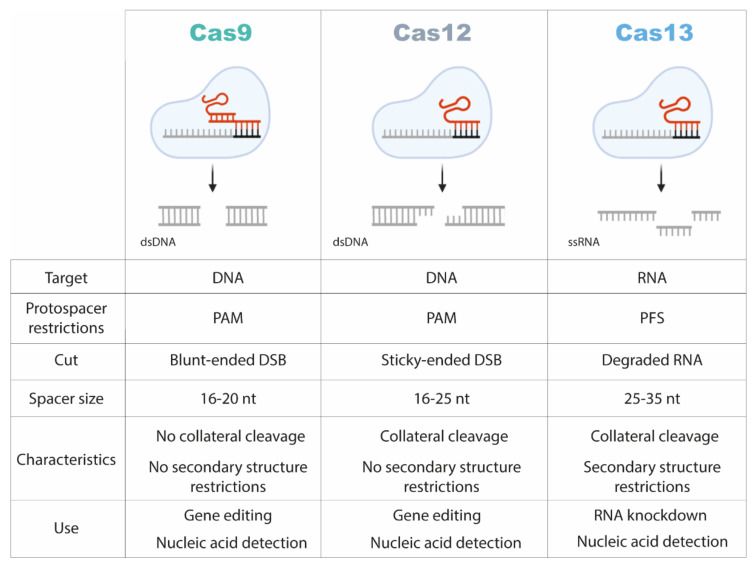
Comparison of the properties of CRISPR/Cas9, /Cas12 and/Cas13 systems. PAM: protospacer adjacent motif; PFS: protospacer flanking site; DSB: double-strand break.

**Figure 3 ijms-23-01757-f003:**
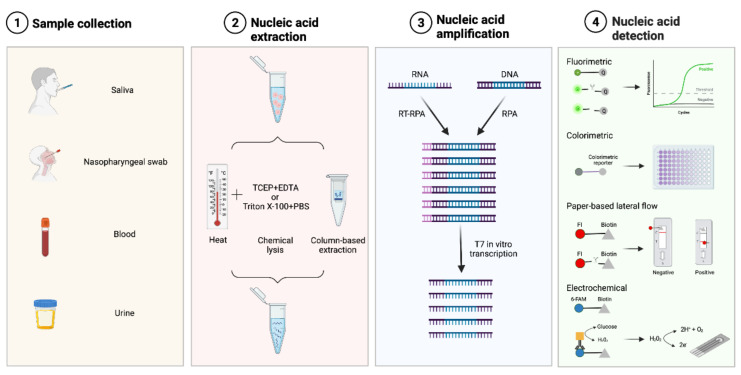
Schematic of the steps required for Cas13-based diagnostics. **1**. Sample can be collected from saliva, nasopharyngeal secretions, blood or urine. **2**. Nucleic acids are extracted using different methods depending on the diagnostic tool. Heat and chemical lysis are combined for a quick extraction; column-based is used for standard extraction. **3**. Nucleic acids can be amplified by different isothermal amplification protocols such as recombinase polymerase amplification (RPA) in DNA samples, or reverse transcription RPA (RT-RPA) in RNA samples, followed by in vitro T7 transcription of the amplified product into RNA. **4**. The activation of the Cas13 enzyme is produced after the binding of the crRNA to the complementary target sequence, triggering collateral cleavage of fluorometric, colorimetric, biotin, or electrochemical reporters.

## Data Availability

Data sharing not applicable.

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
