# Peer review of "CRISPR Approaches for the Diagnosis of Human Diseases"

_ijms, 2022, doi:10.3390/ijms23031757_

Round 1

Reviewer 1 Report

This paper reviews the CRISPR based approaches for human diseases detection. The classification, processing mechanism and applications of CRISPR based methods were introduced. Moreover, the challenges and future prospects of CRIPSR based diagnostics platforms were discussed. However, although this manuscript falls within the aim and scope of this journal, I don’t think this should be published due to lack of sufficient novelty. There are too many reviews about CRISPR-powered diagnostics and this manuscript does not contain additional insights which would justify anther review on this topic. I have to reject this review manuscript in consideration of following aspects:

  • The content of the manuscript is unclear. The title is “CRISPR approaches for the diagnosis of human disease”, however, the big part of the paper was introducing Cas13 family.
  • The sections are differently divided. No connection between different sections. The section 4 was introducing the Cas13 family, while the section 5 was describing type VI crRNA architecture and processing mechanism. The relationship between these two sections without any logic. In addition, in the section 7, Why the content of “qPCR: the gold standard method” was added here? And, thy the section 8 was specificity and sensitivity? I can not see any logic in this manuscript.
  • This manuscript seems to review the CRISPR base diagnostics, but lots of content are unrelated and should be removed. The author should focus on CRISPR platform or Cas13/sgRNA effector mediated detection method.

Reviewer 2 Report

An interesting and informative review of the possibility of using CRISP technologies to diagnose various diseases. 

Round 2

Reviewer 1 Report

ok, the revision can be accepted